# Inspection confirmed mold damage in schools and new use of drugs for airway obstruction: A cohort study

Juha Pekkanen[1,2]*, Martin Täubel[2,3], Lauri Lehtimäki[4,5], Tero Marttila[6], Anne M. Karvonen[2]

1 Department of Public Health, University of Helsinki, Helsinki, Finland, 2 Department of Public Health, Finnish Institute for Health and Welfare, Helsinki, Finland, 3 Department of Civil Engineering, Aalto University, Espoo, Finland, 4 Allergy Centre, Tampere University Hospital, Tampere, Finland, 5 Faculty of Medicine and Health Technology, Tampere University, Tampere, Finland, 6 Faculty of Built Environment, Tampere University, Tampere, Finland

* juha.pekkanen@helsinki.fi

## Abstract

New asthma is considered the most important possible long-term consequence of indoor dampness and mold. However, there are no prospective studies available from schools, there is insufficient evidence on dose-response, and no consensus on how exposure should be assessed. In Finland, visible mold is rare in schools and invasive methods are used to detect mold hidden in building structures.

### Objective

To test if extent and severity of mold damage in school buildings increases the risk of new use of drugs for airway obstruction among students with a dose-response.

### Methods

Extent of mold damage in 110 Finnish primary and secondary school buildings in 2004 was estimated based on all technical inspections done by 2021. New asthma (n = 1,035) and use of drugs for obstructive airway diseases (n = 3,162) among 30,418 students by 2019 was defined based on drug purchases. Multilevel Cox models were adjusted for confounders obtained from Medical Birth Register.

### Results

Extensive mold damage was common. Nine buildings had no or small mold damage, 19 buildings limited, 44 wide and 38 very wide damage in 2004. No association with onset of asthma was observed neither in all students nor in students with previous obstructive airway problems. Among primary school students, there was some suggestion for an association with new use of drugs for obstructive airway diseases, but with highest risk in buildings with limited damage.

**Data availability statement:** Present data cannot be shared publicly on the individual level due to confidentiality reasons and due to the conditions of the permits that we received from Finnish Institute for Health and Welfare (THL/2951/6.02.00/2020, THL/3636/6.02.00/2021 and THL/5583/6.02.00/2022) and from Social Insurance Institution (Kela 120/522/2021 and Kela 13/522/2023) on the use of the register data. Register data from Finnish Institute for Health and Welfare and from Social Insurance Institution, but also from other registers in Finland, can today be applied from a single authority, Findata. Please see https://findata.fi/en/permits/ for instructions on how to apply.

**Funding:** J.P. reports financial support was provided by Research Council of Finland (www.aka.fi), decisions 338679 and 339666. AMK reports financial support from Ministry of Social Affairs and Health (www.stm.fi), decision VN/9508/2021. Open access funded by Helsinki University Library Funders did not play any role in the study design, data collection and analysis, decision to publish, or preparation of the manuscript.

**Competing interests:** The authors have declared that no competing interests exist.

## Conclusions

The results from this unique and large follow-up study suggest that assessment of the extent and severity of mold damage inside building structures in Finnish schools do not identify buildings, in which students are at increased risk of developing asthma.

## Introduction

Moisture damage and mold in homes are considered major environmental risk factors for respiratory symptoms and asthma [1–3]. Although respiratory symptoms and asthma exacerbations have been repeatedly studied in various indoor environments beyond homes, there are no prospective studies examining the association between moisture damage or mold and the development of new asthma in multiple schools or other public buildings [1–4].

There are also many other gaps in our knowledge of the health effects of exposure to moisture damage and mold. Previous studies have largely relied on self-reported assessments of both exposure and health outcomes, which introduces the potential for bias [5]. To address this, more recent research has utilized objective, inspection-based assessments of exposure [6–9]. However, these methods often result in smaller sample sizes. The lack of commonly accepted markers for moisture damage [7–8] or harmful microbial growth [10] further complicates both the assessment process and the comparability of results across studies.

Reviews of existing literature have suggested a possible dose–response relationship between moisture damage or mold and various health effects [7–8]. However, the evidence remains weak, as most studies have used only binary indicators to describe exposure. The lack of detailed dose–response data and standardized exposure markers makes regulatory control of moisture damage and mold challenging. Despite the large number of studies on the health effects of moisture damage and mold, few countries have established binding guidelines. In the United States, ASHRAE has taken initial steps toward developing such guidance [11].

The situation in Finland provides a unique opportunity to address many of these knowledge gaps. In Finland, moisture and mold problems are a subject of active public debate and they cause concern in, e.g., schools [12–13]. Consequently, moisture and mold damage are actively searched for in building investigations and repaired. In Finland's cool and subarctic climate, moisture damage and mold often remain hidden within building structures and visible mold is rare [14]. As a result, detailed invasive inspection methods have been actively developed and implemented since the 1990s to assess the extent of damage in public buildings [15]. This data on exposure in public buildings can be combined with health data using the excellent Finnish national registries.

We therefore conducted a large, registry-based follow-up study examining objectively assessed mold damage in 110 Finnish school buildings. Given that asthma symptoms and current asthma have been extensively studied also in schools and considering that asthma is regarded as the most likely long-term health consequence

of indoor mold exposure, we focused on the development of new asthma and the initiation of medication for obstructive airway diseases among 30,418 primary and secondary school students.

## Methods

### Study design

This prospective follow-up study was designed based on an existing record of all primary and secondary school students in February 2004 in a geographically defined area in Southern Finland. The students in the schools in 2004 were followed up until end of 2019 using record-linkage with National Drug Purchase Register [16] maintained by the Social Insurance Institution (Kela). Confounder data was obtained with record-linkage with Medical Birth Register [17] maintained by Finnish Institute for Health and Welfare (THL). In addition, home address and mortality data were obtained from national registries. The likely extent of mold damage in the school buildings in 2004 was estimated based on retrospective review of all records on building inspections done in the schools between 2002 and 2021.

The study followed the Strengthening the Reporting of Observational Studies in Epidemiology (STROBE) reporting guidelines [18].

### Definition of exposure

Exposure assessment is described and discussed in more detail in the Supplement (S1 Text). In short, we had access to all inspection reports from these primary and secondary school buildings starting from year 2002. Mold damage in the buildings in 2004 was estimated by one of the authors (T.M.) based on retrospective review of all inspection reports done in the buildings between 2002 and end of 2021.

'Small damage' refers to single, local damage with a size of at the most 1–2 m$^2$ in 2004. 'Limited damage' was defined as local mold damage in several locations, but still potentially relevant to indoor exposure in clearly less than one third of the building's floor area. 'Wide damage' was estimated to have possibly affected more than one third of the building's floor area and 'very wide damage' practically the entire building. Typically, in very wide damage students had been moved to another building to avoid exposure or it had been decided to bring forward the complete renovation of the whole building preceded by temporary measures to reduce exposure, like use of air cleaners. A separate category was defined for buildings demolished due to mold damage.

### Definitions of outcomes

We had two outcomes to assess new use of drugs for obstructive airway diseases, one specific for new asthma and another very sensitive one (use of drugs for obstructive airway diseases).

'New asthma' was defined to occur at the first purchase of inhaled corticosteroids, which was followed with further purchases of inhaled corticosteroids so that the amount purchased was for two years on the average at least 100 µg/day of fluticasone propionate equivalent. The amount of fluticasone propionate equivalent inhaled corticosteroids purchased was calculated using conversion factors for different drugs [19].

'New use of drugs for obstructive airway diseases' was defined to occur at the first purchase of any ATC-code R03 drug, which was followed with another purchase of any, not necessarily the same, ATC-code R03 drug within 12 months, but not earlier than two weeks [20]. Biological drugs (R03DX05, R03DX08, R03DX09, R03DX10, R03DX11) were, however, not considered in the classification.

### Definition of confounders

Data on maternal occupation, language and smoking during pregnancy, on caesarean section, birth weight, gestational age, and on older siblings was obtained with record-linkage with Medical Birth Register [17].

Social class was defined based on mother's occupation by dividing it to the following four categories: upper white-collar workers, lower white-collar workers, blue-collar workers and others, and missing. Because mother's social class had a very high proportion of missing data, missing was defined as an own category. Proportion of missing data was 5.5% and 92.7% among primary and secondary school students, respectively. Students with missing data on other confounders but mother's social class were excluded from the final study population (Fig 1). Mother's language was divided into three categories: Finnish, Swedish, and other. Mother's language and social class were analyzed as categorical variables with 3 and 4 levels, respectively. Other confounders were analyzed dichotomized, see Table 1 for cut-off points used.

## Final study population

We included in the study only primary and secondary schools, which had more than 80 students (Fig 1). We did not include schools with several buildings if we were not able to reliably link different grades and students to a specific building. There were four primary schools, where classes up to the 2nd grade and from 3rd to 6th grade were in different buildings, other schools had only one building. We excluded seven buildings, for which we did not have sufficient information to evaluate exposure. This left us with 110 school buildings and 33,418 students with non-missing data on most confounders. There were 88 buildings with primary school students and 32 buildings with secondary school students, i.e., ten buildings had both types of students.

## Statistical methods

Students with asthma before the start of the follow-up were excluded from analyses on development of new asthma. Similarly, students with previous use of drugs for obstructive airway diseases were excluded from the statistical analyses of that outcome. For these exclusions, we utilized data on all drug purchases since birth.

The follow-up period was defined to start when the exposure started, i.e., when the child was assumed to have entered the school building, which he/she attended in February 2004 (S2 Fig). In Finland, preschool starts in the year when the child turns 6 and primary school when the child turns 7. The school year starts in the middle of August, when also all new

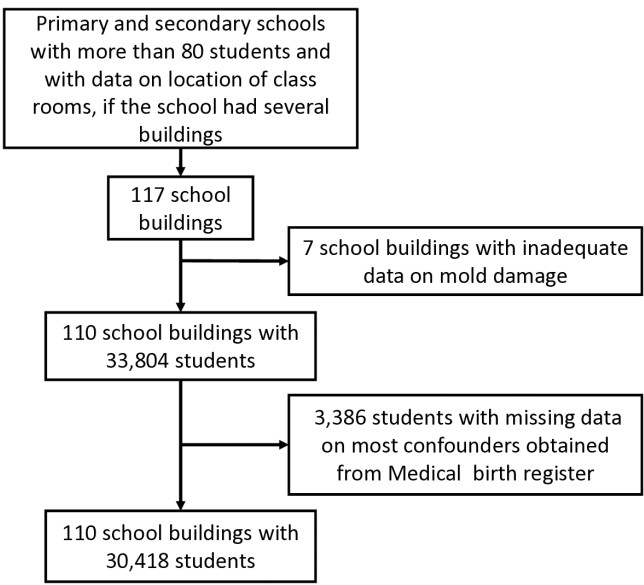

**Fig 1. Flow chart of the study population and buildings.**

**Table 1. Characteristics of the buildings and students by extent of mold damage.**

| | No or small | Limited | Wide | Very wide[a] |
|---|---|---|---|---|
| **Primary school students** | | | | |
| N of buildings[b] | 9 | 14 | 34 | 31 |
| Built in year (mean, range) | 1988 (1913, 2002) | 1948 (1844, 1997) | 1956 (1894, 1999) | 1967 (1905, 1991) |
| N of students per building (mean, range) | 216 (140, 290) | 245 (107, 406) | 249 (59, 560) | 301 (23, 638) |
| N of students | 1943 | 3423 | 8467 | 9341 |
| Age of students (mean±SD) | 9.7±2.3 | 9.9±2.0 | 9.7±2.0 | 9.9±2.0 |
| Girls (%) | 45.7 | 47.2 | 47.8 | 48.1 |
| Maternal smoking during pregnancy[c] (%) | 22.7 | 21.0 | 17.4 | 20.5 |
| Caesarean section (%) | 17.3 | 17.6 | 18.0 | 17.5 |
| Birth weight below 2500g (%) | 5.4 | 4.5 | 4.2 | 4.0 |
| Gestational age<36 weeks (%) | 3.0 | 3.0 | 2.9 | 3.1 |
| No older siblings (%) | 46.3 | 42.3 | 41.8 | 44.1 |
| Mother's social class high (%) | 15.6 | 22.6 | 25.2 | 19.6 |
| Foreign mother tongue (%)[d] | 7.2 | 3.7 | 3.3 | 3.9 |
| **Secondary school students** | | | | |
| N of buildings[b] | 0 | 5 | 15 | 12 |
| Built in year (mean, range) | | 1950 (1880, 1991) | 1956 (1899, 1985) | 1969 (1934, 1990) |
| N of students per building (mean, range) | | 212 (166, 300) | 216 (26, 465) | 245 (50, 421) |
| N of students | | 1061 | 3241 | 2942 |
| Age of students (mean±SD) | | 14.7±1.0 | 14.7±0.9 | 14.8±1.0 |
| Girls (%) | | 46.1 | 49.7 | 47.9 |
| Maternal smoking during pregnancy[c] (%) | | 12.6 | 11.5 | 16.6 |
| Caesarean section (%) | | 15.7 | 17.0 | 16.4 |
| Birth weight below 2500g (%) | | 3.8 | 3.5 | 4.5 |
| Gestational age<36 weeks (%) | | 2.2 | 2.8 | 3.0 |
| No older siblings (%) | | 49.8 | 50.6 | 52.8 |
| Mother's social class high (%) | | 2.3 | 2.0 | 1.5 |
| Foreign mother tongue (%)[d] | | 0.6 | 0.6 | 0.7 |

[a]Also buildings demolished due to mold damage.

[b]There were 10 buildings with both primary and secondary school students.

[c]Smoking during pregnancy also after the first trimester.

[d]Mother's language not Finnish or Swedish

students start school. Therefore, the follow-up period, e.g., for a student on the 2nd grade in February 2004 was coded to have started on August 15, 2002, if there was no preschool in the same building, and on August 15, 2001, if there was a preschool in the same building. The length of the follow-up was on average (in sum) 18.8 (549,818) years and 18.0 (475,214) years in the analyses on asthma and on use of drugs for airway obstructive diseases, respectively.

Risk of new events was analyzed with multilevel multivariate Cox regression with building entered as a random variable. Subjects were censored at the time of death, moving abroad, or 31.12.2019, whichever came first. The proportionality assumption was checked with inspection of the Kaplan-Meier curves. All analyses were done using SAS Enterprise Guide 8.3 software (© 2019–2020, SAS Institute Inc., Cary, NC, USA).

As sensitivity analyses, we analyzed separately primary school students whose home address had not changed during the years they were in primary school (S3 Table). In these analyses we excluded also those four primary schools, where classes were in two different buildings.

We performed separate analyses dividing the follow-up into the periods during and after the time the student attended the school (S4 Table) and on the risk of asthma among students with previous use of drugs for obstructive airway diseases (Table 5).

## Ethics

This study was approved by the Research Ethics Committee of the Faculty of Medicine, University of Helsinki. The Finnish Institute for Health and Welfare (THL/2951/6.02.00/2020, THL/3636/6.02.00/2021 and THL/5583/6.02.00/2022) and Social Insurance Institution (Kela 120/522/2021 and Kela 13/522/2023) gave permission to access the data from the national health registers. Different registers were assessed between 1.12.2021–1.12.2022. The authors did not have access to information that could identify individual participants during or after data collection. This is a register-based study in which the study subjects were not contacted.

## Results

The number of school buildings with no mold damage was two and with small mold damages seven. All those buildings had only primary school students (Table 1). Due to small numbers, these two categories were combined for statistical analyses. Also, there were only five buildings that had been demolished due to mold damage (S5 Fig), also having only primary school students. The unadjusted risk of new asthma and new use of drugs for obstructive airway disease in the demolished buildings (3.5% and 12.0%) were like those in buildings with very wide damage (3.3% and 12.8%, respectively). Therefore, demolished buildings were combined into the very wide damage category.

In the final analyses, nine buildings were coded to have had no or small mold damage, 19 buildings limited, 44 wide (>1/3 of building area) and 38 very wide damage in 2004. Among the 101 buildings with more than small damage, damage was observed before 2010 in 56 (55%) buildings, between 2010–2014 in 30 (30%) buildings, and between 2015–2021 in 15 (15%) buildings.

Primary school buildings with no or small damages were newer (Fig 2) and had somewhat less students (Table 1) compared to school buildings with more extensive mold damage. Mothers of the students from the schools with no or small damage were twice as likely to have a foreign background and had somewhat lower social class. Otherwise, the levels of confounders were similar in different categories of mold damage in buildings having both primary and secondary students (Table 1).

No significant associations were observed between extent of damage and development of new asthma when primary or secondary schools were analyzed separately or together (Table 2). There was some suggestion of a higher risk in primary school buildings with limited or wide damage, but not with very wide damage, as compared to no or small damage. This was better visible in the analyses on development of new use of drugs for obstructive airway diseases (Table 3), where the comparison between no or small damage with limited damage just reached statistical significance (p = 0.0463) in the fully adjusted model.

Year of construction of the building was not associated with either new asthma or new use of drugs for airway obstruction (Table 4). The risks also differed little by year when the damage had been observed. Different adjustments had very little effect on the main conclusions (Tables 2–4).

Conclusions were similar from the analyses limited to primary school students, whose home address did not change during primary school and after excluding those four primary schools, where classes were in two different buildings (S3 Table). However, there was no longer any suggestion for a lower risk of new asthma in the buildings with no or small

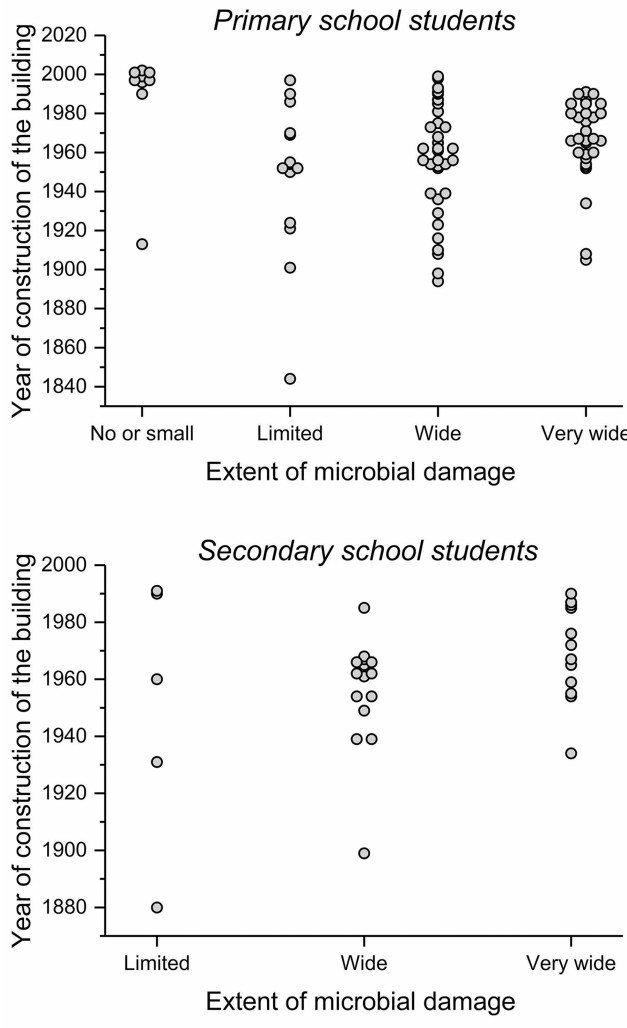

**Fig 2. Year of construction of the buildings by extent of mold damage. Each building is marked with a dot.**

damage. Instead, the results showed a non-significant decreasing risk with increasing extent of mold exposure. Associations with new use of drugs for airway obstruction were slightly stronger. These analyses excluded 1,157 (8.3%) and 1,008 (8.1%) students from the analyses on new asthma and on new use of drugs for obstruction airway diseases, respectively.

Analyses among primary school students (S4 Table) suggest that the increased risk of new use of drugs for obstruction airway diseases mainly occurred while the student was still attending the school, as compared to after leaving the school. Despite the higher hazard ratios, they did not reach statistical significance ($p < 0.05$).

Additional analyses were done among a potentially susceptible subgroup, i.e., those 2881 students who fulfilled the definition of new use of drugs for obstructive airway diseases used in the present study, but not the definition of new asthma, before start of the follow-up (Table 5). While these students had a clearly higher risk of developing asthma, the odds ratios between extent of damage and development of new asthma were very similar compared to all the students free of asthma at the start of the follow-up (Table 2).

**Table 2. Association of mold damage in the school building with development of new asthma using two different adjustments, separately for primary and secondary school students and for all students combined.**

| Mold damage | N of buildings | students at risk | n (%) of cases | Model 1[a] hazard ratios (95% CI) | Model 2[b] hazard ratios (95% CI) |
|---|---|---|---|---|---|
| *Primary school students* | | | | | |
| No or small | 9 | 1870 | 56 (3.0%) | 1 | 1 |
| Limited | 14 | 3288 | 126 (3.8%) | 1.24 (0.87, 1.77) | 1.14 (0.73, 1.77) |
| Wide | 34 | 8098 | 300 (3.7%) | 1.21 (0.88, 1.66) | 1.09 (0.72, 1.66) |
| Very wide | 31 | 8954 | 284 (3.2%) | 1.03 (0.75, 1.42) | 0.93 (0.60, 1.44) |
| *Secondary school students* | | | | | |
| Limited | 5 | 1030 | 33 (3.2%) | 1 | 1 |
| Wide | 15 | 3127 | 124 (4.0%) | 1.16 (0.77, 1.74) | 1.00 (0.62, 1.60) |
| Very wide | 12 | 2861 | 112 (3.9%) | 1.09 (0.72, 1.65) | 0.90 (0.53, 1.55) |
| *All students*[c] | | | | | |
| No, small or limited | 28 | 6188 | 215 (3.5%) | 1 | 1 |
| Wide | 44 | 11225 | 424 (3.8%) | 1.06 (0.88, 1.28) | 0.99 (0.80, 1.23) |
| Very wide | 38 | 11815 | 396 (3.4%) | 0.93 (0.77, 1.12) | 0.88 (0.71, 1.11) |

Asthma defined based on 2-year use of inhaled corticosteroids.

[a]Model 1 adjusted for student's age and sex.

[b]Model 2 adjusted in addition for year of construction, year of detection of moisture damage, mother's social class, mother's language, maternal smoking during pregnancy, caesarean section, birth weight below 2500g, gestational age < 36 weeks, and not having older siblings.

[c]Models for all students adjusted also for a binary variable for primary vs secondary school student.

**Table 3. Association of mold damage in the school building with new use of drugs for obstructive airways diseases using two different adjustments, separately for primary and secondary school students and for all students combined.**

| Mold damage | N of buildings | students at risk | n (%) of cases | Model 1[a] hazard ratios (95% CI) | Model 2[b] hazard ratios (95% CI) |
|---|---|---|---|---|---|
| *Primary school students* | | | | | |
| No or small | 9 | 1643 | 166 (10.1%) | 1 | 1 |
| Limited | 14 | 2936 | 370 (12.6%) | 1.24 (1.03, 1.48) | 1.24 (1.00, 1.54) |
| Wide | 34 | 7277 | 890 (12.2%) | 1.19 (1.01, 1.40) | 1.19 (0.97, 1.46) |
| Very wide | 31 | 7984 | 956 (12.0%) | 1.16 (0.99, 1.37) | 1.17 (0.94, 1.45) |
| *Secondary school students* | | | | | |
| Limited | 5 | 940 | 101 (10.7%) | 1 | 1 |
| Wide | 15 | 2900 | 350 (12.1%) | 1.02 (0.81, 1.29) | 0.91 (0.67, 1.25) |
| Very wide | 12 | 2668 | 329 (12.3%) | 0.98 (0.78, 1.25) | 0.87 (0.62, 1.24) |
| *All students*[c] | | | | | |
| No, small or limited | 28 | 5519 | 637 (11.5%) | 1 | 1 |
| Wide | 44 | 10177 | 1240 (12.2%) | 1.03 (0.94, 1.14) | 0.97 (0.87, 1.08) |
| Very wide | 38 | 10652 | 1285 (12.1%) | 1.01 (0.91, 1.11) | 0.94 (0.84, 1.06) |

Use of drugs for obstructive airways diseases defined based on two prescriptions for any R03 drug within one year.

[a]Model 1 adjusted for student's age and sex.

[b]Model 2 adjusted in addition for year of construction, year of detection of moisture damage, mother's social class, mother's language, maternal smoking during pregnancy, caesarean section, birth weight below 2500g, gestational age < 36 weeks, and not having older siblings.

[c]Models for all students adjusted also for a binary variable for primary vs secondary student

**Table 4. Effect of different mutual adjustments of three building characteristics on multivariate adjusted hazard ratios (HR) for development of new asthma and use of drugs for obstructive airways disease. Primary and secondary school students combined. P-values below 0.1 marked with a footnote.**

| | N of buildings | students at risk | n of new cases | HR[a] (No mutual adjustment) | HR[b] (Bivariate model) | HR[c] (Bivariate model) | HR[d] (Mutual adjustment) |
|---|---|---|---|---|---|---|---|
| *Development of new asthma* | | | | | | | |
| Year of construction | | | | | | | |
| Before1940 | 22 | 5235 | 198 | 1 | 1 | | 1 |
| 1940- 1959 | 22 | 5717 | 222 | 1.06 | 1.08 | | 1.07 |
| 1960-1969 | 23 | 6572 | 222 | 0.92 | 0.95 | | 0.91 |
| 1970-1989 | 23 | 6830 | 230 | 0.95 | 0.99 | | 0.96 |
| 1990- | 20 | 4874 | 163 | 0.95 | 0.95 | | 1.06 |
| Mold damage | | | | | | | |
| No, small or limited | 28 | 6188 | 215 | 1 | 1 | 1 | 1 |
| Wide | 44 | 11225 | 424 | 1.04 | 1.03 | 0.98 | 0.99 |
| Very wide | 38 | 11815 | 396 | 0.93 | 0.92 | 0.86 | 0.88 |
| Year of detection of mold damage | | | | | | | |
| No or small | 9 | 1870 | 56 | 1 | | 1 | 1 |
| before 2009 | 56 | 14460 | 537 | 1.15 | | 1.24 | 1.32 |
| 2010−14 | 30 | 9482 | 333 | 1.09 | | 1.17 | 1.20 |
| after 2015 | 15 | 3416 | 109 | 1.00 | | 1.05 | 1.06 |
| *Development of new use of drugs for obstructive airways disease* | | | | | | | |
| Year of construction | | | | | | | |
| Before1940 | 22 | 4820 | 583 | 1 | 1 | | 1 |
| 1940- 1959 | 22 | 5142 | 638 | 1.07 | 1.07 | | 1.08 |
| 1960-1969 | 23 | 5906 | 705 | 1.01 | 1.02 | | 1.04 |
| 1970-1989 | 23 | 6166 | 737 | 1.01 | 1.02 | | 1.02 |
| 1990- | 20 | 4314 | 499 | 0.99 | 0.99 | | 1.04 |
| Mold damage | | | | | | | |
| No, small or limited | 28 | 5519 | 637 | 1 | 1 | 1 | 1 |
| Wide | 44 | 10177 | 1240 | 1.03 | 1.01 | 0.97 | 0.97 |
| Very wide | 38 | 10652 | 1285 | 1.00 | 0.99 | 0.95 | 0.94 |
| Year of detection of mold damage | | | | | | | |
| No or small | 9 | 1643 | 166 | 1 | | 1 | 1 |
| before 2009 | 56 | 13098 | 1581 | 1.16 | | 1.21[e] | 1.21 |
| 2010−14 | 30 | 8565 | 1055 | 1.18[e] | | 1.22[e] | 1.24[e] |
| after 2015 | 15 | 3042 | 360 | 1.15 | | 1.18 | 1.18 |

All hazard ratios (HR) also adjusted for student's age and sex, primary/secondary student, mother's social class, mother's language, maternal smoking during pregnancy, caesarean section, birth weight below 2500g, gestational age<36 weeks, and not having older siblings

[a]No mutual adjustement: All three building characteristics in separate models

[b]First bivariate model: Year of construction and mold damage in the same model

[c]Second bivariate model: Mold damage and year of detection of mold damage in the same model

[d]Full mutual adjustement: All three building characteristics in the same model

[e]p-values between 0.036 and 0.05

**Table 5. Association of mold damage in the school building with development of new asthma among students with previous obstructive airway problems. Analyses includes those 2881 students who fulfilled the definition for new use of drugs for obstructive airway diseases used in the present study, but not the definition for new asthma, before start of the follow-up. Separately for primary and secondary school students and for all students combined.**

| Mold damage | N of buildings | students at risk | n (%) of cases | Model 1[a] hazard ratios (95% CI) | Model 2[b] hazard ratios (95% CI) |
|---|---|---|---|---|---|
| *Primary school students* | | | | | |
| No or small | 9 | 227 | 21 (9.3%) | 1 | 1 |
| Limited | 14 | 353 | 38 (10.8%) | 1.13 (0.64, 1.99) | 1.24 (0.62, 2.50) |
| Wide | 34 | 821 | 79 (9.6%) | 1.01 (0.61, 1.69) | 1.10 (0.57, 2.14) |
| Very wide | 31 | 970 | 81 (8.4%) | 0.85 (0.51, 1.41) | 0.91 (0.45, 1.84) |
| *Secondary school students* | | | | | |
| Limited | 5 | 90 | 13 (14.4%) | 1 | 1 |
| Wide | 15 | 227 | 35 (15.4%) | 1.08 (0.51, 2.27) | 1.12 (0.38, 3.33) |
| Very wide | 12 | 193 | 30 (15.5%) | 1.05 (0.49, 2.26) | 1.12 (0.33, 3.76) |
| *All students[c]* | | | | | |
| No, small or limited | 28 | 670 | 72 (10.7%) | 1 | 1 |
| Wide | 44 | 1048 | 114 (10.9%) | 0.96 (0.70, 1.31) | 0.95 (0.66, 1.37) |
| Very wide | 38 | 1163 | 111 (9.5%) | 0.84 (0.62, 1.15) | 0.90 (0.61, 1.32) |

Use of drugs for obstructive airways diseases defined based on two prescriptions for any R03 drug within one year. Asthma defined based on 2-year use of inhaled corticosteroids.

[a]Model 1 adjusted for student's age and sex

[b]Model 2 adjusted in addition for year of construction, year of detection of moisture damage, mother's social class, mother's language, maternal smoking during pregnancy, caesarean section, birth weight below 2500g, gestational age < 36 weeks, and not having older siblings.

[c]Models for all students adjusted also for a binary variable for primary vs secondary school student.

## Discussion

This 15-year follow-up study is one of the largest follow-up studies on moisture damage and mold with objective assessment of exposure [1–4,6] and the first follow-up study done in multiple schools or other public buildings [3–4]. Results show that exposure to extensive inspection confirmed mold damage was common in the studied Finnish schools and several of the buildings were even urgently taken out of use or demolished due to mold damage. Despite this, no association was observed with future risk of asthma neither among all students nor among students at high risk of asthma due to previous obstructive airway problems. The results from this unique and large follow-up study with objective assessment of exposure suggest that assessment of the extent and severity of mold damage inside building structures may not serve the purpose of identifying school buildings, in which students have an increased risk of developing asthma. An alternative explanation for our findings is that the increases in asthma risks are too small to detect even in large follow-up studies using building-level assessment of exposure.

The method used in the present study to assess mold damage differs from previous prospective studies done in homes, which have shown an association between moisture damage and mold and development of new asthma [1–3,6,21], also in Finland [22–24]. Most of these past studies have been based on self-reports or short walk-through inspections for visible signs of moisture or mold growth or smell in homes [1–4,6]. In Finland's cool and subarctic climate, visible mold is rare [14,25] and moisture damage and mold are typically hidden inside the structures of the building, especially in structures with design errors in the construction, so called risk structures [15, 26, 27]. Therefore, detailed invasive methods have been developed to assess the extent of such damage [15,26,28]. Unrepaired microbial growth in buildings leading to possible exposure is considered a potential health hazard in the Finnish legislation [29] and actively intervened on.

Mold damage hidden in structures requires air connection to the indoor air to create exposure for occupants of the building. Such damage is therefore different from observations of visible mold for example in the living areas of a home, where exposure via indoor air can be directly assumed. The degree of exposure to a hidden damage with air connection to occupied spaces in a building will always vary depending on the extent of air leaks, pressure difference between indoor and outdoor, and other factors, which is why it can often be difficult to determine the degree of air connection and even harder to evaluate the amount of exposure. Air tightness of the buildings varies a lot between building types and between individual buildings even within the same building type. Based on average air leakage rates, precast concrete blocks of flats are clearly more airtight than wooden detached houses [30]. Air leaks are nevertheless common, especially in older building stock [27]. In a recent study of school buildings located in the same area as the present study, regular air leaks were detected in one third and partial air leaks in practically all the studied school buildings [25]. However, when trying to explain the discrepancy in the observed associations between the present study and past studies, lower exposure due to missing or lower air connection from the mold damage to indoor air is probably a partial explanation.

Students spend much less time at school compared to at home, which also reduces exposure in the schools compared to homes. Schools in Finland also practically always operate with mechanically forced ventilation that is effective in mitigating potential airborne exposures, including that from mold damage. Both factors likely partly explain the lack of associations observed in the present study. There are also several differences in the materials and methods used in construction of homes and schools. E.g., detached and semidetached homes are mostly made of wood in Finland, whereas schools are more often made of concrete or masonry bricks. Wood is more sensitive to microbial growth under moisture stress [14] and these differences may also affect the microbiological indoor air quality in schools [31]. The importance of these potential differences is, however, unclear as the evidence on the association of microbiological characteristics of moisture damage on asthma risk is still scarce [10]. Another potential explanation is that exposure to moisture and mold happens later in life at school than at home and it has been suggested that first few years of life are an especially sensitive period in the development of asthma [32]. However, similar relative risks for the association between residential moisture and mold and new asthma have been reported in adult studies [3,33] as for children [3,21], which makes this a more unlikely explanation.

Past meta-analyses based on studies done in homes [3,21] have shown that the relative risks for development of new asthma ranges largely between 1.3 to 1.7, depending on the dampness and mold indicator considered. Given the shorter exposure duration and better ventilation in schools, one would expect smaller relative risks for development of asthma in schools than in homes. The present results are consistent with this assumption. The present study can naturally not rule out a small increased risk of developing asthma also in schools with mold damage hidden inside structures, but our findings suggests that if there is an increased risk, the relative risk is likely clearly smaller than 1.3. Such a small relative risk can only be detected in large cohort studies and would probably require also a more detailed personal assessment of exposure, not just a building-level exposure assessment as in the present study.

While most past studies have used dichotomous variables to describe exposure to moisture damage and mold, recent reviews of studies with multi-level moisture damage and mold exposure estimates have found some suggestions for a dose-response between moisture damage and mold and various health effects [7–8]. However, studies on development of asthma, which have used multi-categorical markers of exposure to moisture and mold are still very few and the evidence for dose-response relationship is sparse [9,23,24,33–35]. Such information would be crucial for standard setting. For this reason, we made a dedicated effort in our study into multi-level exposure estimates to assess a possible dose-response, but none was detected.

We had two outcomes to assess new use of drugs for airway obstruction, one specific for asthma, i.e., regular use of inhaled corticosteroids, and one more sensitive, i.e., at least two purchases of any drugs for obstructive airway diseases, used in previous studies on asthma [20,36]. The observed incidence of asthma in the present study was consistent with the incidence of asthma estimated for Finland [37], supporting the reliability of the outcome definition we used. It has also

been shown that data on medication use obtained from registries is a valid indicator of asthma [38]. However, we did not have data that would have enabled us to separate different phenotypes of asthma, like allergic asthma. As not all cases of new asthma become diagnosed, and some may not receive the recommended regular medication, we also used a more sensitive outcome in the analyses. This turned out to be very sensitive, as two-thirds of the students who fulfilled the criterion for the more sensitive outcome, did not develop asthma based on the more specific criterion. As obstructive airway diseases other than asthma are rare among children, the more sensitive outcome likely captured mainly more mild, transient and undefined airway symptoms unrelated to any specific disease. This is suggested also by the somewhat higher hazard ratios for the more sensitive outcome during the time the student attended the school (S4 Table).

Nine (10%) of the primary school buildings had no or only small damages. In these nine buildings the risk of new use of drugs for obstructive airway diseases was lower than in other primary school buildings, but there was no dose-response, and no association was observed with new asthma. The finding may be at least partly due to exposure misclassification and residual confounding. These nine buildings with no or small damage were clearly newer than other buildings (Fig 2), with seven out of the nine being built after year 1995. In the present study, buildings built after 1995 were assumed to have no mold damage if none had been detected, even if no technical inspections had been done. Also, mothers of the students in these schools were more likely from a foreign background and of lower social class, possible related to the fact that families living in the new neighborhoods built in Helsinki in the 1990's were more often of lower social class and often from foreign background [39]. Social background may affect health care utilization and purchase of drugs for obstructive airway diseases. We were able to adjust our analyses only for crude markers of culture and socioeconomic status, which leaves room for residual confounding. In addition, age of the building, which in previous studies has been shown to be associated with risk of mold damage [14,40] was not associated with future risk of either new asthma or use of drugs for obstructive airway diseases in the present study (Table 4). Taken together, we do not think that the present analyses support a causal association between extent of mold damage in schools and risk of new use of drugs for obstructive airway diseases.

The main strengths of the present study are the large sample size, complete and long follow-up, and objective assessment of both exposure and outcome. We used information from nationwide registries, which reduces the risk of recall bias, because the medication data is collected from registries instead of questionnaires or interviews. On the other hand, we had no information about adherence to medication, only information about the quantities of medication purchased. We also had available several confounders for asthma, but socioeconomic status was estimated based on mother's occupation only. As the study was based on registries only, we had no information on many typical determinants of asthma, like pets or mold exposure at home. Given the small effect of adjusting for the available confounders, we consider it likely that also adjusting for these missing determinants would have made little effect on the results.

We estimated students' exposure on the building-level, like earlier cross-sectional studies in schools, which have shown associations of moisture and mold with respiratory symptoms and asthma [4]. Also, there were on average only little more than 200 students in the school buildings of the present study, so the buildings were mostly small. As the association between moisture and mold with new asthma may well be weaker in schools than in homes, the association may be hard to detect without a more detailed assessment of the students' individual exposure (see S1 Text).

One would expect that the risk of new asthma would be higher during exposure, i.e., during the time the student spent in the damaged school building, compared to the time after leaving the damaged building, as on-going mold exposure may increase respiratory symptoms and thereby trigger detection of new asthma among student with pre-existing, but still not diagnosed asthma. Also, the follow-up up to the age of 31 years for oldest secondary school students could potentially dilute the observed associations. We therefore conducted a sensitivity analysis comparing asthma risk during exposure and after leaving school. There was some suggestion for stronger associations during exposure, but no significant associations or suggestions for dose-response were detected.

Predisposing factors are important for some adverse effects of mould, but few studies have observed predisposing factors for new asthma [2]. Individual studies have shown stronger associations between mould and new asthma among those with previous atopic sensitization [24] and with previous rhinosinusitis [41]. In the present study, we did not have data on these specific predisposing factors, but we performed analyses on a potentially susceptible symptomatic subgroup of students who had used drugs for obstructive airway diseases before start of the follow-up. These students had a clearly higher risk of developing new asthma, but the strength of the association between extent of damage and development of new asthma was very similar to that observed among all students.

The small number of undamaged buildings is in contrast to epidemiological studies done in homes, where the challenge has typically been the small number of homes with widespread, confirmed damage [9,23,24]. However, previous studies on prevalence of mold damage in public buildings in Finland using more similar methods as the present study have observed high prevalences of damage [14–15]. Despite the small number of undamaged buildings, we expected to find a difference between buildings with limited damage (clearly less than 1/3 of the building's floor area) and very wide damage affecting practically the whole building and leading to acute measures to reduce exposure or to demolishing the buildings, but none was detected.

The present study is one of the largest follow-up studies on moisture damage and mold using data from detailed building inspections and the first follow-up study done in a larger number of public buildings, rather than homes. Widespread, technically confirmed mold damage, mainly hidden inside building structures, was common in the studied Finnish school buildings. However, no association was seen with risk of developing new asthma. As the present study is the first of its kind, the results need to be confirmed in new prospective studies. Like previous cross-sectional studies in schools [4], exposure was estimated on the building level, which also reflects the way buildings are currently ranked based on their estimated risks and repair needs. Taken together, the results from this unique and large follow-up study suggest that assessment of the extent and severity of mold damage inside building structures do not identify school buildings with students at increased risk of developing asthma.

## Supporting information

**S1 Text. Methods and discussion of exposure assessment.**
(DOCX)

**S2 Fig. Graphical presentation of the design.**
(DOCX)

**S3 Table. Association of mold damage with new asthma and new use of drugs for obstructive airways diseases. Analyses limited to primary school students, whose home address did not change during primary school.**
(DOCX)

**S4 Table. Association of mold damage with new asthma and new use of drugs for obstructive airways diseases, separating the follow-up into periods during and after the school years.**
(DOCX)

**S5 Fig. Year of construction of the school building by extent of mold damage.**
(DOCX)

## Acknowledgments

We thank Asko Vepsäläinen and Merja Korajoki for performing the statistical analyses, and Jukka Lahdensivu and Miia Pitkäranta for constructive comments. Open access funded by Helsinki University Library.

## Author contributions

**Conceptualization:** Juha Pekkanen.

**Formal analysis:** Juha Pekkanen.

**Funding acquisition:** Juha Pekkanen, Martin Täubel, Anne M. Karvonen.

**Investigation:** Juha Pekkanen, Lauri Lehtimäki, Tero Marttila, Anne M. Karvonen.

**Supervision:** Juha Pekkanen, Anne M. Karvonen.

**Writing – original draft:** Juha Pekkanen.

**Writing – review & editing:** Martin Täubel, Lauri Lehtimäki, Tero Marttila, Anne M. Karvonen.

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
