## [Decision Letter · Decision Letter 0]

12 Jun 2025

PONE-D-25-22528Inspection confirmed mold damage in schools and new use of drugs for airway obstruction: A cohort studyPLOS ONE

Dear Dr. Pekkanen,

Thank you for submitting your manuscript to PLOS ONE. After careful consideration, we feel that it has merit but does not fully meet PLOS ONE’s publication criteria as it currently stands. Therefore, we invite you to submit a revised version of the manuscript that addresses the points raised during the review process. In particular, both reviewers were concerned with the unspecificity of the outcome for mould exposure that shoukld be discussed in more detail. Also, to the correlation with maternal SES a few more statements should be  dedicated in the discussion. Please ensure that your decision is justified on PLOS ONE’s publication criteria  and not, for example, on novelty or perceived impact.

We look forward to receiving your revised manuscript.

Kind regards,

Thomas Behrens

Academic Editor

PLOS ONE

Additional Editor Comments (if provided):

Reviewers' comments:

Reviewer's Responses to Questions

**Comments to the Author**

1. Is the manuscript technically sound, and do the data support the conclusions?

Reviewer #1: No

Reviewer #2: Partly

2. Has the statistical analysis been performed appropriately and rigorously? 

Reviewer #1: N/A

Reviewer #2: Yes

3. Have the authors made all data underlying the findings in their manuscript fully available?

Reviewer #1: Yes

Reviewer #2: Yes

4. Is the manuscript presented in an intelligible fashion and written in standard English?

Reviewer #1: No

Reviewer #2: Yes

5. Review Comments to the Author

Reviewer #1: The aim of this study was to examine whether the extent and severity of mould damage in school buildings increases the risk of newly initiated medication use due to airway obstruction among students, and whether a dose-response relationship can be established.

Mould infestation in buildings can lead to various health impairments in sensitive and particularly predisposed individuals – ranging from unpleasant odours and irritations to allergies and asthmatic symptoms. However, asthmatic symptoms may have diverse causes and are not solely dependent on mould infestation in school buildings. A direct association between mould infestation and increased medication use is therefore not immediately evident, making the study’s finding – that there is no association between mould infestation and students’ medication use – is not surprising.

While mould exposure was assessed in detail and various confounding factors were taken into account, no information was available on the students’ health complaints. It remains unclear why maternal socioeconomic status was correlated with mould exposure.

Although the data from the registry study and the exposure assessment were extensive, the information is insufficient to answer the study’s research question. In particular, the students’ health status is not adequately documented. Data should be included on whether, among other factors, there is an early childhood atopic predisposition.

Reviewer #2: My opinion is that the outcome variables (new asthma or new obstructive airway disease) are too nonspecific for mold exposure and a specific part of discussion should be dedicated to that part. Also we know from many studies that a low dose of allergen is allergenic and a high dose exposure induces tolerance. This should be also discussed in detail. This two points are important when interpreting the obtained results.

6. PLOS authors have the option to publish the peer review history of their article (what does this mean? ). If published, this will include your full peer review and any attached files.

**Do you want your identity to be public for this peer review?** For information about this choice, including consent withdrawal, please see our Privacy Policy .

Reviewer #1: No

Reviewer #2: No

---

## [Author Response · Author response to Decision Letter 1]

1 Sep 2025

Plos One

Dear editors,

Please find below our responses to the reviewers comments.

The editor commented: “In particular, both reviewers were concerned with the unspecificity of the outcome for mould exposure that should be discussed in more detail. Also, to the correlation with maternal SES a few more statements should be dedicated in the discussion.”

We have now carefully addressed both of these concerns.

In addition to the below changes requested by the reviewers, we have made following changes to the manuscript:

1. In previous analyses, we used mother’s nationality as a confounder, but this variable had a lot of missing values. As maternal SES is potentially an important confounder, we have now replaced mother’s nationality with mother’s language and rerun all analyses. This resulted, however, in only minor changes in the second decimal of the odds ratios.

2. While redoing the analyses, we noticed that there was a mistake in the footnotes of all tables except tables 1 and 4: model 1 is adjusted only for age and sex. This is now corrected.

On behalf of all of the authors

Helsinki 23.8.2025

Juha Pekkanen

Reviewers' comments:

Reviewer's Responses to Questions

Comments to the Author

1. Is the manuscript technically sound, and do the data support the conclusions?

Reviewer #1: No

Reviewer #2: Partly

2. Has the statistical analysis been performed appropriately and rigorously?

Reviewer #1: N/A

Reviewer #2: Yes

3. Have the authors made all data underlying the findings in their manuscript fully available?

Reviewer #1: Yes

Reviewer #2: Yes

4. Is the manuscript presented in an intelligible fashion and written in standard English?

Reviewer #1: No

Reviewer #2: Yes

5. Review Comments to the Author

Reviewer #1: The aim of this study was to examine whether the extent and severity of mould damage in school buildings increases the risk of newly initiated medication use due to airway obstruction among students, and whether a dose-response relationship can be established.

Mould infestation in buildings can lead to various health impairments in sensitive and particularly predisposed individuals – ranging from unpleasant odours and irritations to allergies and asthmatic symptoms. However, asthmatic symptoms may have diverse causes and are not solely dependent on mould infestation in school buildings. A direct association between mould infestation and increased medication use is therefore not immediately evident, making the study’s finding – that there is no association between mould infestation and students’ medication use – is not surprising.

Response 1: Thank you for this comment. It is true that asthma and asthmatic symptoms have many causes, like practically all medical problems. The challenge in epidemiological studies is to control those causes, which confound the association under study, and to avoid other biases. We discuss in length that the control for confounders and assessment of exposure was not perfect, but were pleased with the reviewer’s comment on that ‘mould exposure was assessed in detail and various confounding factors were taken into account’ (see next comment). It should also be noted that previous evidence on the association between mould damage and development of new asthmatic problems comes exactly from the same type of studies as the present one (see references 1-6). Therefore, we did expect to see at least a weak direct association between mould in schools and development of new asthma, which however was not the case, despite a very large study population.

While mould exposure was assessed in detail and various confounding factors were taken into account, no information was available on the students’ health complaints.

Response 2: This comment refers to two possible points, which are also raised in other comments: first, that better information on students’ health complaints at baseline could help to identify a susceptible subgroup of students (see response 2.1); and second, that better information on students’ health complaints at follow-up would be needed to answer the research question of the present study (see response 2.2)

Response 2.1: It is true that predisposing factors are important for some adverse effects of mould, but only very few studies have observed predisposing factors for new asthma associated with mould damage (reference 2). One study has shown stronger associations between mould and new asthma among those with previous atopic sensitization (see reference 24) and one with previous rhinosinusitis (see new reference 41). A new paragraph has been added on this on lines 392-399.

As we now point out in the discussion (line 395), we did not have information of students’ atopic sensitization or past atopic eczema or rhinosinusitis, which could potentially be used to identity sensitive subjects. We also did not have data on self-reported symptoms, but we have now done additional analyses on a subgroup of 2881 students with previous respiratory complaints (based on medication use), but who had not yet developed asthma before the start of the follow-up. The results are very similar to those among all the students. The result has been added as new table 5 and to the Abstract, Methods (lines 160-161) and Discussion (lines 278 and 397-399)

Response 2.2: It is true that better information of students’ health complaints at follow-up would be needed to study many potential outcomes related to mould damage, like atopic sensitization. However, the present study’s research question was, if mould damage in schools increases the risk of new asthma or asthmatic problems. This research question was chosen, as new asthma is considered the most important possible long-term consequence of indoor dampness and mold (see references 1-3).

As we now point out in the Discussion, the definitions used in the present study are taken from previous studies on asthma (line 340, reference 20 and the new reference 36). It has also been earlier shown that data on medication use obtained from registries is a valid indicator of asthma (line 342-343, new reference 38). We now point out in the Discussion as a weakness that we did not have data to separate different phenotypes of asthma, like allergic asthma (343-344). As previously, we point out the specificity of the used definition for asthma and the results for another, less specific definition of obstructive airway problems (lines 338-351).

It remains unclear why maternal socioeconomic status was correlated with mould exposure.

Response 3: It is unclear, why such a correlation exists in our data, but it may be related to the fact that families living in the new neighborhoods built in Helsinki in the 1990’s were more often of lower social class and also were more often from foreign background (see new reference 39). This has now been pointed out in the Discussion (lines 359-361).

Although the data from the registry study and the exposure assessment were extensive, the information is insufficient to answer the study’s research question. In particular, the students’ health status is not adequately documented. Data should be included on whether, among other factors, there is an early childhood atopic predisposition.

Response 4: Thank you for the comments. This is discussed in response 2.1 above.

Reviewer #2: My opinion is that the outcome variables (new asthma or new obstructive airway disease) are too nonspecific for mold exposure and a specific part of discussion should be dedicated to that part.

Response 5: We appreciate this comment and have discussed this issue in response 2.2

Also we know from many studies that a low dose of allergen is allergenic and a high dose exposure induces tolerance. This should be also discussed in detail. This two points are important when interpreting the obtained results.

Response 6: The present study’s research question was, if mould damage in schools increases the risk of new asthma or asthmatic problems. Mold exposure is only weakly associated with allergic sensitization (reference 2) and it is hypothesized that the pathogenetic mechanism between mold damage and development of asthma is related to enhancement of inflammatory reactions in the airways, rather that development of sensitization (reference 2). Also, we are not aware of previous evidence suggesting that high dose exposure to moisture damage and indoor mould could induce tolerance, while low dose would cause increased risk of asthma and sensitization. We now point out in the Discussion as a weakness that we did not have data to separate different phenotypes of asthma, like allergic asthma (line 343-344), however, we propose not to discuss in more detail the mechanisms of allergic sensitization and tolerance.

---

## [Decision Letter · Decision Letter 1]

15 Sep 2025

Inspection confirmed mold damage in schools and new use of drugs for airway obstruction: A cohort study

PONE-D-25-22528R1

Dear Dr. Pekkanen,

We’re pleased to inform you that your manuscript has been judged scientifically suitable for publication and will be formally accepted for publication once it meets all outstanding technical requirements.

Kind regards,

Thomas Behrens

Academic Editor

PLOS ONE

**Comments to the Author**

1. If the authors have adequately addressed your comments raised in a previous round of review and you feel that this manuscript is now acceptable for publication, you may indicate that here to bypass the “Comments to the Author” section, enter your conflict of interest statement in the “Confidential to Editor” section, and submit your "Accept" recommendation.

Reviewer #1: All comments have been addressed

Reviewer #2: All comments have been addressed

2. Is the manuscript technically sound, and do the data support the conclusions?

Reviewer #1: Partly

Reviewer #2: Yes

3. Has the statistical analysis been performed appropriately and rigorously? 

Reviewer #1: N/A

Reviewer #2: Yes

4. Have the authors made all data underlying the findings in their manuscript fully available?

Reviewer #1: Yes

Reviewer #2: Yes

5. Is the manuscript presented in an intelligible fashion and written in standard English?

Reviewer #1: Yes

Reviewer #2: Yes

6. Review Comments to the Author

Reviewer #1: Thank you for the addition of further detailed information. Some issues are now more clear and enhanced the quality of the manuscript

Reviewer #2: The authors have answered all raised questions by reviewers in a proper and understandable way. The changes are visible in the text in the part with track changes.

7. PLOS authors have the option to publish the peer review history of their article (what does this mean? ). If published, this will include your full peer review and any attached files.

**Do you want your identity to be public for this peer review?** For information about this choice, including consent withdrawal, please see our Privacy Policy .

Reviewer #1: No

Reviewer #2: No

---

## [Editor Report · Acceptance letter]

PONE-D-25-22528R1

PLOS ONE

Dear Dr. Pekkanen,

I'm pleased to inform you that your manuscript has been deemed suitable for publication in PLOS ONE. Congratulations! Your manuscript is now being handed over to our production team.

Kind regards,

on behalf of

Prof. Thomas Behrens

Academic Editor

PLOS ONE